# Ecological memory of recurrent drought modifies soil processes via changes in soil microbial community

Alberto Canarini ⬡ [1]✉, Hannes Schmidt ⬡ [1], Lucia Fuchslueger[1], Victoria Martin[1], Craig W. Herbold[1], David Zezula[1], Philipp Gündler[1], Roland Hasibeder[2], Marina Jecmenica[1], Michael Bahn ⬡ [2] & Andreas Richter ⬡ [1]✉

Climate change is altering the frequency and severity of drought events. Recent evidence indicates that drought may produce legacy effects on soil microbial communities. However, it is unclear whether precedent drought events lead to ecological memory formation, i.e., the capacity of past events to influence current ecosystem response trajectories. Here, we utilize a long-term field experiment in a mountain grassland in central Austria with an experimental layout comparing 10 years of recurrent drought events to a single drought event and ambient conditions. We show that recurrent droughts increase the dissimilarity of microbial communities compared to control and single drought events, and enhance soil multifunctionality during drought (calculated via measurements of potential enzymatic activities, soil nutrients, microbial biomass stoichiometry and belowground net primary productivity). Our results indicate that soil microbial community composition changes in concert with its functioning, with consequences for soil processes. The formation of ecological memory in soil under recurrent drought may enhance the resilience of ecosystem functioning against future drought events.

[1] Centre for Microbiology and Environmental Systems Science, University of Vienna, Vienna, Austria. [2] Department of Ecology, University of Innsbruck, Innsbruck, Austria. ✉email: alberto.canarini@hotmail.it; andreas.richter@univie.ac.at

The present functioning of ecosystems is determined by the temporal sequence of events that shapes interactions within communities. Yet, when trying to understand current ecosystem dynamics, history is an often-neglected aspect, even though highly relevant, especially in the light of rising climatic variability brought about by global change. In this context, emerging theoretical frameworks and recent evidence point at the important role of ecological memory[1], defined as 'the capacity of past states or experiences to influence present or future responses of a community'[2]. Indeed, the increase in frequency of a disturbance can create a memory of that disturbance, modifying the structure and organisation of ecological interactions within an ecosystem[3]. As global climatic change increases the frequency of extreme events, the temporal dynamic of ecosystem responses becomes a vital information to predict climate change consequences across temporal and spatial scales. For example, microorganisms and microbially mediated processes dominating biogeochemical cycles[4] can be severely affected by drought events, which may lead to substantial alterations of soil carbon (C) and nutrient cycling[5]. Globally, the temporal variability of precipitation will likely become larger than ever[6], which may lead to a higher frequency of drought events. The implications of increased drought frequency on soil greenhouse gas emissions and plant nutrient availability, and thus for ecosystem services as a whole[7], can be severe. Understanding if soil community composition can shift in response to changing conditions[8] and how this can affect ecosystem dynamics is, therefore, crucial to better predict the future alterations of biogeochemical cycles.

A prerequisite to create the memory of a disturbance such as drought is the presence of material or information legacies[9]. While material legacy refers to biotic or abiotic structures of a system (e.g. seed dispersal, soil pH, etc.), information legacy represents species life-history traits of a community. Field studies on legacy effects of drought on the soil microbial community are scarce, which has led many conclusions to be drawn from laboratory incubations. Furthermore, drought is a generic term referring to water shortage, which can either be a chronic or a short but intense precipitation reduction. Previous studies have shown that both types of drought can cause changes to the soil microbial community structure[10–13] and that chronic precipitation reduction showed no legacy effects after rewetting of soil[12,14]. On the contrary, when drought was simulated as an intense stress event, it had significant long-lasting legacy effects on the microbial community, altering soil–plant feedback with changes in plant growth, nutrient acquisition, belowground C allocation and ecosystem functioning[11,13,15–17]. Such legacy effects were found at both during a consecutive drought event and after drought was terminated[13,15,16], although it is not known yet how long these effects can last. Some studies have also investigated long-term effects of chronic drought in the field, but used laboratory incubation to disentangle short-term vs long-term effects[12,18], potentially missing complex field dynamics. Results from field experiments comparing long-term recurrent drought events to single drought events are still lacking.

Here, we report the investigation of in situ drought effects of a system subjected to 10 years of recurrent drought events compared to a single drought event on the soil microbial community composition and the processes it mediates. The overarching goal of this work was to assess the formation of an ecological memory in soil microbial processes and the presence of legacies in the soil microbial community (in terms of composition, physiological parameters and stoichiometry). We hypothesized that: (i) recurrent drought events would cause larger shifts in the microbial community composition, compared to a single drought event, and (ii) recurrent drought events would lead to an ecological memory response of microbially mediated ecosystem processes.

This study is part of a long-term field drought experiment in a mountain grassland in Austria[19]. At this field site, drought is repeatedly implemented as an intense stress event during the growing season by completely excluding precipitation for about 2 consecutive months. In 2017, we compared grassland plots exposed to summer drought for 10 consecutive years with newly exposed ones and ambient controls. This allows us to disentangle drought responses, driven by microbial community shifts, to long-term recurrent drought from short-term physiological responses. Microbially driven carbon (C), nitrogen (N), phosphorous (P) and sulfur (S) cycling is assessed using potential enzymatic activity of 10 different enzymes. In addition, we analyse microbial biomass stoichiometry, microbial growth and carbon use efficiency (CUE), and community composition based on phospholipid fatty acids (PLFAs) and amplicon sequencing approaches. Additional soil data available at this site for plots subjected to 1, 2, 3, 4 and 5 years of recurrent summer drought events (data obtained in 2011 and 2012) is also analysed to understand the temporal aspects leading to an ecological memory development. To our knowledge, this is the first study that provides an integrative analysis of an ecological memory formation caused by recurrent drought events on the soil biogeochemical cycles mediated by the soil microbial community. We further describe insights into microbial community response to long-term drought, and we consider the implications for soil functionality in response to climate change. Specifically, our work shows two key findings: (i) recurrent drought strengthens shifts in the composition of microbial communities; and (ii) soil multi-functionality is buffered in response to drought by previous recurrent drought events. We further suggest that microbial community shifts and changes in the microbially mediated cycling of carbon and nutrients likely reflect community-dependent strategies to acclimate to drought, and likely controlling soil functions.

## Results

**Drought does not affect measured edaphic factors but modifies available C and N.** Soil pH and total soil C, N and P contents (and their respective ratios) did not differ significantly between treatments (Supplementary Table 1). The experiment reduced soil water content throughout the experimental period (Supplementary Fig. 1, Supplementary Table 1 and Supplementary Fig. 2), with no differences between plots subjected to 10 years of recurrent drought events (10-year treatment) and plots exposed to a single drought event (1-year treatment). Similar results were obtained in the previous campaigns that were carried out in 2011 and 2012 (Supplementary Figs. 1 and 2). Independent of the number of drought events, extractable organic C (EOC) concentrations were higher compared to ambient controls. Among the measured available N pools, only $NH_4^+$ showed significant differences between treatments, where values followed the order (1 year > 10 years > control). Extractable inorganic and organic P (EIP and EOP) were not significantly affected by drought (Supplementary Table 1).

**Ten years of recurrent drought events cause legacy effects on microbial biomass stoichiometry and community composition.** The magnitude and direction of responses of the microbial biomass and its stoichiometry differed significantly between the 1-year and 10-year treatments. A similar trend was found for all the measured variables where the 1-year treatment diverged from control values while the 10-year treatment had similar values to control. This trend was found significant for microbial biomass N (MBN, $p = 0.025$) and microbial biomass P (MBP, $p = 0.046$), whereas microbial biomass C (MBC) was not significant (Fig. 1, top panels).

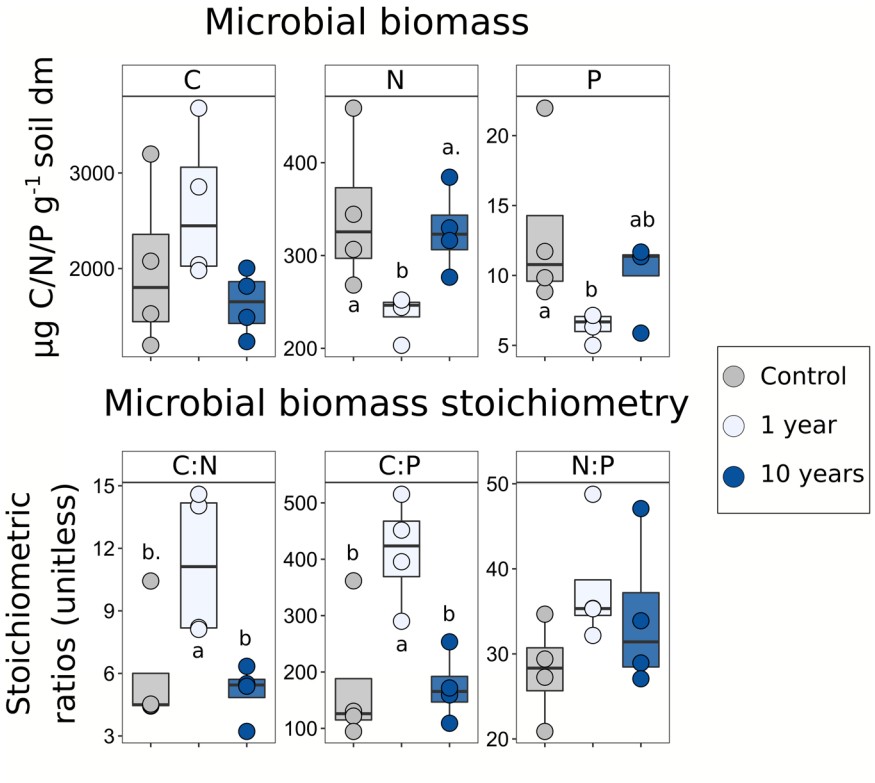

**Fig. 1 Drought effects on microbial biomass C, N and P.** Colour indicates soil treatment (control = grey, 1 year = light blue, 10 years = blue). The top graphs show microbial biomass carbon (MBC), nitrogen (MBN) and phosphorous (MBP). MBC was not significantly affected by drought ($F$-value: 2.02, $p = 0.18$), while MBN and MBP were ($F$-value: 5.7, $p = 0.04$, $n = 4$; $F$-value: 4.44, $p = 0.06$, $n = 4$). The same pattern was present in all the three values, where control and the 10-year treatment had similar values and the 1-year treatment was different. The bottom graphs show the stoichiometric ratios of microbial biomass. The ratio of C-to-N and C-to-P show similar results, where the 1-year treatment was significantly different from both control and 10 years (respectively, $F$-value: 5.72, $p = 0.0249$, $n = 4$; $F$-value: 8.06, $p = 0.0098$, $n = 4$). The ratio of N-to-P was not statistically affected by drought, and the 10-year treatment values ranged between the 1 year and control. Letters represent results from two-sided Tukey HSD post hoc test (a point next to letter represents a $p$-value on the 0.05 threshold). Box centre line represents median, box limits the upper and lower quartiles, whiskers the 1.5x interquartile range, while separated points represent outliers. The sample size '$n$' represents biologically independent samples.

MBC and MBN were also analysed for the previous campaigns (2011 and 2012), with no clear drought effect up to 5 years of recurrent drought events (Supplementary Fig. 3). The ratios of C:N and C:P in microbial biomass significantly increased ($p = 0.016$ and $p = 0.003$, respectively) in response to the single drought event (1 year) but not after 10 years of recurrent drought, when compared to ambient plots (Fig. 1, bottom panels). The microbial N:P ratio followed a similar, although not significant, pattern.

Although total microbial biomass (sum of all PLFA biomarkers), remained constant in all drought treatments compared to controls (Supplementary Table 2), drought had a clear impact on the microbial community composition (PERMANOVA: $R^2 = 0.38$; $p < 0.05$; Supplementary Fig. 4), mostly driven by the 10-year treatment. Indeed, the 10-year drought clearly separated from the control, whereas the microbial community in the 1-year treatment overlapped with both control and 10-year treatment. Results from previous campaigns show that drought did not result in a significant difference between drought treatments and control in the 1-, 2- and 5-year treatments (Supplementary Fig. 5). The separation towards control plots was mostly driven by biomarkers belonging to arbuscular mycorrhizal fungi (AMF), Gram-positive and Gram-negative bacteria (e.g. a17:0, i17:0, 16:1ω5), while shifts in drought plots were related to fungal biomarkers (e.g. 18:2ω6,9). When individual biomarkers were used to estimate the biomass of main microbial groups,

significant effects of drought were found only in Gram-negative and Gram-positive bacteria ($p = 0.011$ and $p = 0.016$; Supplementary Table 2) and in protozoa ($p = 0.001$; Supplementary Table 2), with a significant decrease. The NLFA (neutral lipid fatty acid) biomarker 16:1ω5 was assessed as an indicator of AMF biomass and no statistical difference was found between treatments and control (Supplementary Table 2).

A more detailed picture of the microbial community response to drought, based on amplicon sequencing, revealed that 10 years of recurrent drought events had caused a significant microbial community shift, clearly separating it from single drought event (1 year) and control (Fig. 2, top panels), whereas there was no clear separation between control and 1 year. We analysed the V4 region of the SSU rRNA genes (for bacteria and archaea), the ITS1 region (for fungi in general) and AMF-ITS2 rDNA (for AMF). For all, bacteria, archaea, fungi and AMF, a significant effect of drought was found (PERMANOVA test: $p = 0.014$, $p = 0.01$, $p = 0.015$, respectively). To confirm the results from ITS1, the ITS2 region was also analysed showing very similar results (Supplementary Fig. 6). Differential abundance analysis further confirmed (Fig. 2, bottom panels) that the 10-year treatment led to a much higher number of differentially abundant taxa (compared to the control) than the 1-year treatment. The 10-year treatment reduced the relative abundances compared to controls of the groups: Acidobacteria, Bacteroidetes,

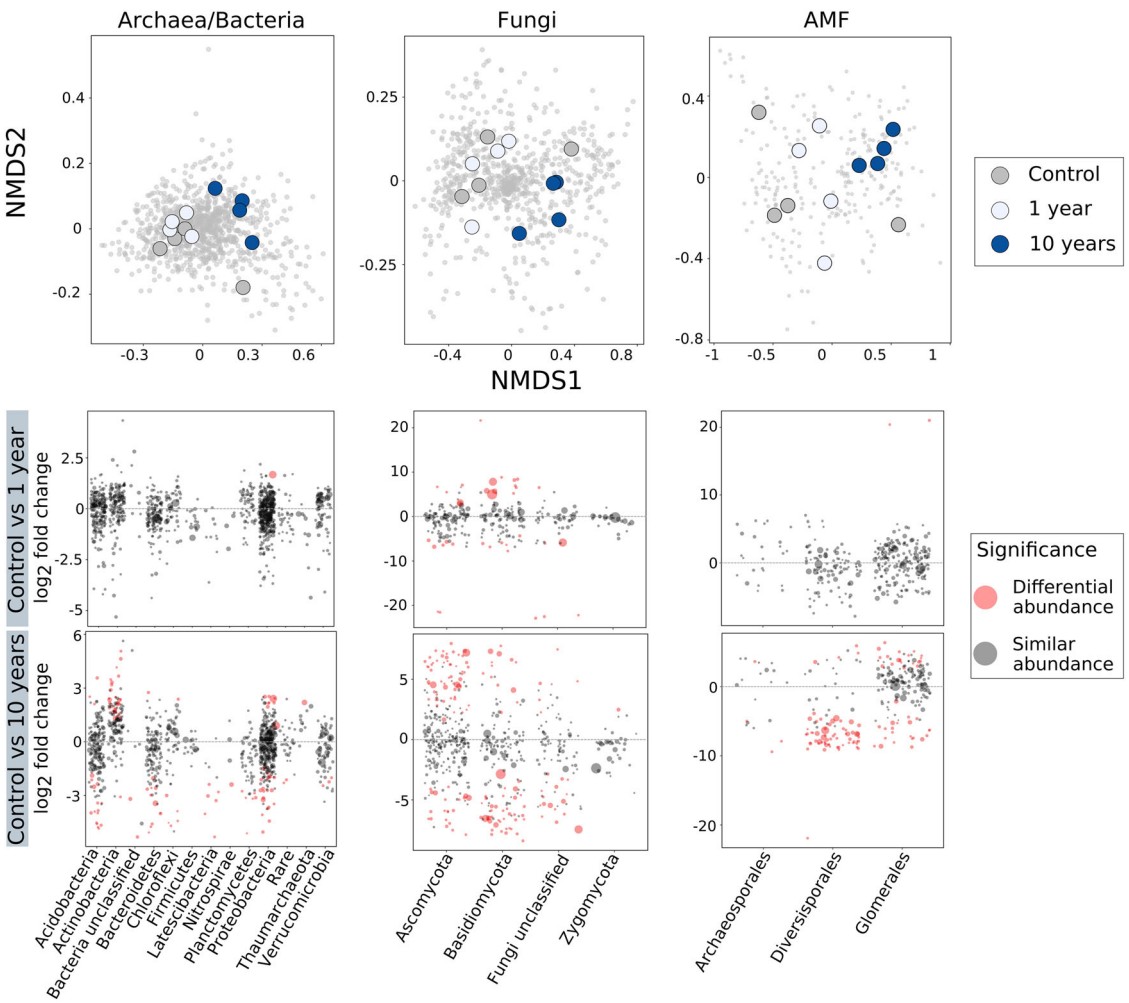

**Fig. 2 Drought effects on microbial community composition assessed by amplicon sequencing.** Colour indicates soil treatment (control = grey, 1 year = light blue, 10 years = blue). On top, graph NMDS representing archaea/bacteria, fungi and AMF (arbuscular mycorrhizal fungi). All the NMDS plots show the similar patterns, with an effect of drought (respectively, $p = 0.014$, $p = 0.01$, $p = 0.015$; PERMANOVA results, $n = 4$) and with a clear separation of the 10-year treatment from 1 year and control. The bottom graphs show results from the differential abundance analysis for control vs 1 year (top row) and control vs 10 years (bottom row). Circles in red represent taxa that had a significant differential abundance compared to the control, while in grey taxa that were not significant. Similarly to the NMDS plots, control vs 1 year results show little difference (only in the fungal community of Basidiomycota), while control vs 10 years shows strong differences at all levels (most affected groups being: Acidobacteria, Actinobacteria, Proteobacteria, Ascomycota, Basidiomycota, Diversisporales and Glomerales).

Basidiomycota, fungi unclassified and Diversisporales, whereas Actinobacteria, Ascomycota and Glomerales showed higher relative abundance and Proteobacteria showed both higher and lower relative abundance (Supplementary Figs. 7, 8, 9, 10). No clear patterns were evident in the α-diversity indices (Richness, Chao1, ACE, Shannon and Inverse Simpson index) between treatments (Supplementary Fig. 11). We further analysed Legacy Response Groups (LRGs) and show the relative abundance of the main taxonomic groups that were either enriched or depleted in the 10-year vs the 1-year treatment (Supplementary Fig. 12). We specifically selected these two treatments to disentangle long-term vs short-term effects of drought. We found that most phyla (or orders) contributed at the same time to both types of LRGs (enriched and depleted groups). The main relative abundance taxa in the enriched LRGs belonged to the phyla Latescibacteria, Proteobacteria, Acidobacteria and Thaumarchaeota, within the bacterial and archeal community. Ascomycota followed by Basidiomycota represented the major enriched LRGs in the fungal community and the order Glomerales in the arbuscular mycorrhizae community.

**Ecological memory formation in the microbially mediated cycling of C and nutrients.** To analyse whether repeated droughts changed the potential of the microbial community functions and nutrient acquisition strategies in response to drought, we analysed a suite of extracellular enzymatic activities categorized as C-, N-, P-, S-cycle-related enzymes (see Supplementary Table 3). The enzymes were classified as C-related, when the enzymes were involved in the cycling of molecules that did not contain N, P or S, or related to one of the nutrients when involved in the depolymerization of molecules that contain N, P or S. It should be noted that, as all organic molecules contain C, it is impossible to completely separate the acquisition of C from the acquisition of nutrients contained in organic molecules (as in N-, P- and S-related enzymes, Supplementary Table 3). The drought treatments caused enzyme-specific responses (Fig. 3, top panels). C-cycle-related enzymes showed consistent patterns where the 10-year treatment had values always similar to the control, whereas the 1-year treatment was always different from both (although only β-xylosidase showed a significant effect of drought; $p = 0.016$). Similarly, in the N-cycle enzymes the 1-year

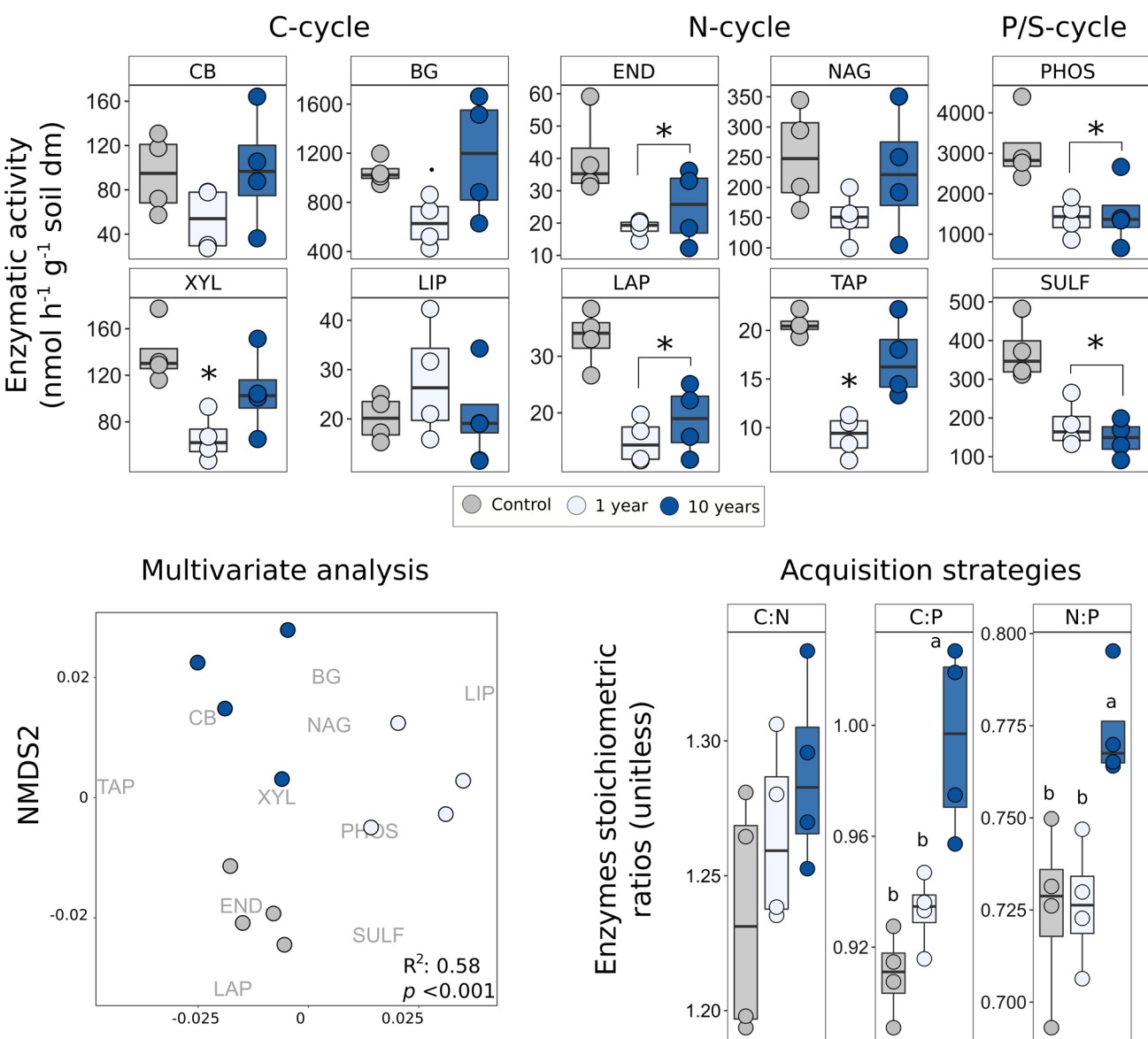

**Fig. 3 Effects of drought on the potential enzymatic activity in soil.** Colour indicates soil treatment (control = grey, 1 year = light blue, 10 years = blue). Top graphs show the potential enzymatic activities of all measured enzymes, grouped for C-, N-, and P- and S-cycle-related enzymes (for full description of enzymes see Supplementary Table 1). C-cycle enzymes all show a similar trend, where 10 years have similar values to the control, and the 1 year is always different (only statistically significant in XYL: $F$-value = 6.68, $p$ = 0.016, $n$ = 4; and partially significant in BG: $F$-value = 3.2, $p$ = 0.089, $n$ = 4). N-cycle enzymes show different responses of the 10-year treatment, where, in two enzymes both drought treatments show a decrease (END: $F$-value = 4.9, $p$ = 0.036, $n$ = 4; LAP: $F$-value = 14.58, $p$ = 0.0015, $n$ = 4), and in the other two only the 1-year treatment shows decreased values, while 10 years show values similar to control (only significant in TAP: $F$-value = 18.79, $p$ < 0.001, $n$ = 4). P- and S-cycle enzymes show the same trend where both drought plots are significantly different from the control (PHOS: $F$-value = 6.53, $p$ = 0.018, $n$ = 4; SULF: $F$-value = 14.75, $p$ = 0.0014, $n$ = 4). Stars represent significant differences with the Tukey HSD post hoc test while a dot represents partially significant values ($p$ < 0.1). Bottom left panel shows the NMDS of potential enzymatic activities for the year 2017, with a clear separation between all treatments ($R^2$ = 0.58, $p$ = 0.0003, $n$ = 4, stress = 0.13). The bottom right panel represents acquisition strategies (stoichiometric ratios of the potential enzymatic activities), where a similar trend was found for all the ratios, although only statistically significant in C:P and N:P (respectively, $F$-value = 14.8, $p$ = 0.001, $n$ = 4; $F$-value = 8.6, $p$ = 0.008, $n$ = 4). Letters represent results from two-sided Tukey HSD post hoc test. Box centre line represents median, box limits the upper and lower quartiles, whiskers the 1.5x interquartile range, while separated points represent outliers. The sample size '$n$' represents biologically independent samples.

drought treatment caused stronger reductions than the 10-year drought treatment when compared to the control (significant effects of drought were found for three enzymes: Endochitinase, Leucine aminopeptidase and Tyrosine-aminopeptidase, $p$ = 0.03, $p$ = 0.001 and $p$ = 0.0006, respectively, Fig. 4). In contrast, S- and P-cycle enzymes showed a significant decrease in activity to both

1-year and 10-year treatments, compared to the control (Fig. 4). Enzymatic activity was also expressed per MBC, which led to very similar conclusions (Supplementary Fig. 13). Similar trends were also found in the previous sampling campaigns (2011 and 2012), for which we observed a statistically significant linear trend of Cellobiosidase towards control values with increasing years of

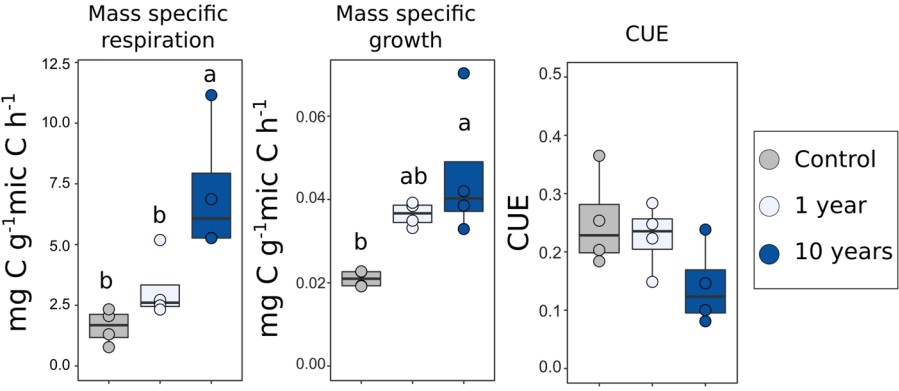

**Fig. 4 Physiological parameters measured upon rewetting.** Colour indicates soil treatment (control = grey, 1 year = light blue, 10 years = blue). Drought had a significant effect on mass-specific respiration and growth rates (respectively, F-value = 9.67, p = 0.006, n = 4 and F-value = 14.55, p = 0.0015, n = 4), whereas it had no significant effect on CUE. The 10-year drought was significantly different from 1-year drought in mass-specific respiration rates. Letters represent results from two-sided Tukey HSD post hoc test. Box centre line represents median, box limits the upper and lower quartiles, whiskers the 1.5x interquartile range, while separated points represent outliers. The sample size 'n' represents biologically independent samples.

drought (Supplementary Fig. 14), whereas Phosphatase, Endo-chitinase and Leucine aminopeptidase did not show a relationship with number of years of drought.

When enzymes were analysed in a multivariate analysis (non-metric multidimensional scaling, NMDS), drought had an overall significant effect across all potential enzymatic activities (PER-MANOVA test p < 0.001; Fig. 3, bottom left panel), where responses to the 1 and 10 years of recurrent drought were clearly separated from each other as well as from the control. Treatment separation corresponded to specific enzymes, where control was related to the Endochitinase and Leucine aminopeptidase, the 10 years was related to Cellobiosidase, β-Glucosidase and Exochiti-nase, and the 1 year to Phosphatase and Lipase. In previous campaigns (2011 and 2012) we also observed a separation of treatments from control plots (although marginally significant), with no clear separation within the different drought treatments of 1, 2, 3, 4 and 5 years of recurrent droughts (Supplementary Fig. 14, bottom graphs).

We analysed the nutrient acquisition strategies of the overall community, calculated as stoichiometric ratio of enzymes. The C:N:P acquisition ratio is an integral feature to describe soil microbial community functions, linking environmental nutrient availability to the C:N:P stoichiometry of microbial biomass[20–22]. We found that 10 years of recurrent drought increased C:P and N:P acquisition ratios (p = 0.001 and p = 0.008, respectively) and a similar trend was also found for C:N (although not significant), whereas acquisition strategies were not affected by the 1-year treatment and remained similar to controls (Fig. 3, bottom right panel).

**Shifts in microbial physiology and belowground multi-functionality in response to drought.** We measured parameters related to microbial physiology (growth, respiration and CUE). Due to methodological reasons, it was necessary to add water to measure these parameters and therefore they represent the physiological rewetting response of the microbial communities subjected to drought. We found an increase in mass-specific respiration rates and growth rates (p = 0.006 and p = 0.017, respectively), but not in the CUE (Fig. 4). The 10-year drought led to average values of all parameters different from control and 1-year drought (Fig. 4), with lower CUE values and higher respiration and mass-specific growth rates, although only mass-specific growth rates were statistically significant.

To obtain a substrate-independent measure of the effects of drought on microbial function and physiology, we predicted the

functional potential of the soil bacterial community based on 16S rRNA gene sequencing using PICRUSt2[23]. We specifically aimed at differentiating the response of the first (1 year) to the tenth (10 years) recurrent drought, in order to (i) disentangle shifts due to previous drought history from simple responses to drought, and also because (ii) there were very little differences between communities in control and 1-year treatment (see Fig. 2). We found that the abundance of major group of functions, such as amino acid metabolism, membrane transport, metabolism of cofactors and vitamins, and cell growth and death differed between the 1-year and 10-year treatments (Fig. 5, upper left graph). However, discussing broad classes of functions may be misleading as they represent multiple genes whose expression was concomitantly decreased and increased (Fig. 5, upper middle graph). For example, while we found no overall change in the total number of genes related to carbohydrates metabolism, hundreds of functions were both enriched and depleted (Fig. 5, middle panel). Therefore, we further explored individual gene responses, especially for processes known to be involved in the improvement of drought tolerance. We found that the number of genes related to the production of osmolytes (glycine betaine, glutamate, proline and choline), including enzymes such as betB, betI, gbsA and proA were enriched. A gene inducible via osmotic stress encoding a membrane transporter for K+ uptake (kdp) was found to be decreased in the 10-year drought; while the genes trkG and trlH, which are also involved in K+ uptake but are constitutive genes, were enriched. Other genes that are responsive to osmotic stress were found to be enriched, including ompC and ompF which express porins that facilitate the non-specific diffusion of small (≤500 Da) hydrophilic molecules[24]. We also found an enrichment of genes connected to the capsular or extracellular polysaccharides (EPS) metabolism (alg7, pslB and wbpW)[25,26], although some were also reduced (e.g. pgaB). Moreover, we observed an enrichment in genes involved in peptidoglycan biosynthesis, like the murA and murB genes[27]. These constitute a shield for bacterial cells from abiotic stresses and help absorbing water[28]. Drought-tolerant bacteria also respond to low water activity by increasing the proportion of negatively charged phospholipids, including phosphatidylglycerol lipids, and we found that bacterial community in the 10-year treatment had higher relative abundance of gene encoding for the phosphatidylglycerophosphatase (pgpA gene). Lastly, ROS sca-vengers were observed to be enriched, like genes for enzymes belonging to the superoxide dismutase (gene SOD2) or catalase families. We also found evidence that bacterial communities may

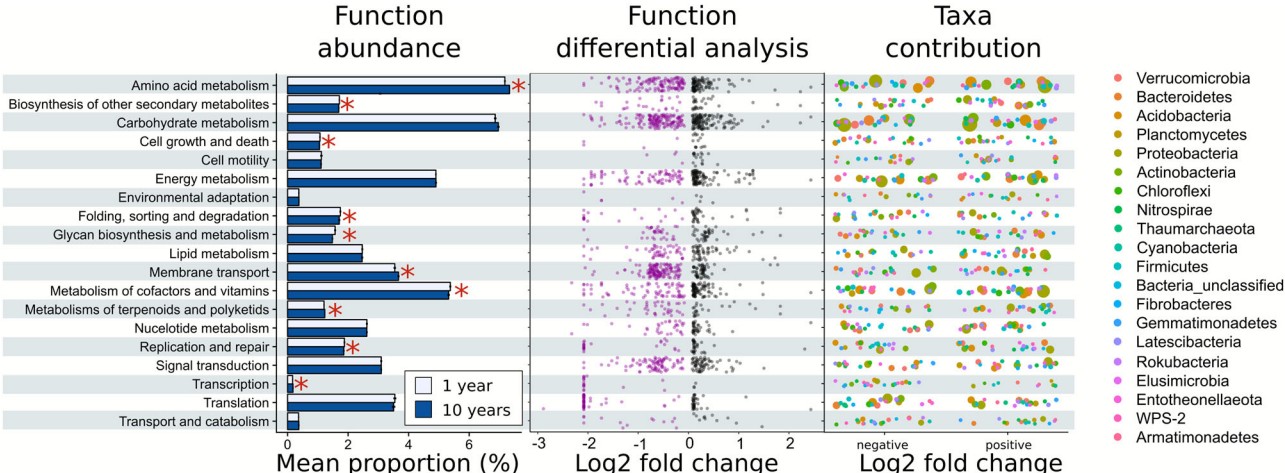

**Fig. 5 Potential functional capabilities and acquisition strategies of soil microbial communities.** The three graphs show results from PICRUSt2 comparing 1-year to 10-year drought. This revealed potential enrichment or depletion (respectively, positive and negative Log2 fold changes) of many functional traits of the microbial community related to important functions in response to drought history. The left graph shows the mean proportion of level 2 KEGG Orthologies within the bacterial community for the 1-year and 10-year treatments. Colour indicates soil treatment (1 year = light blue, 10 years = blue). Red asterisks indicate significant differential abundances of the overall group. The middle graphs represent the number of functions that were enriched (black circles) or depleted (purple circles) using DESeq2 (respectively, positive and negative Log2 fold changes). The right graph represents the microbial groups (bubble size represents bigger or smaller frequency of ASVs in each group of function within each phylum) contributing to enriched or depleted functions (respectively, positive or negative). Individual phylum colour are described in the figure.

have increased their investment in resource acquisition traits to degrade more chemically complex and diverse substrates (such as the enrichment of genes expressing amylases, chitinases and enzymes related to lignin-degradation such as *vdh*), at the expense of pathways related to more simple compounds (glucose, fructose, mannose, etc.). We further used PICRUSt2 outputs to predict the contribution of amplicon sequence variants (ASVs) to individual functional categories. We observed that different microbial members of individual phyla contributed concomitantly in a positive or negative way to each predicted function.

Conclusions drawn from PICRUSt2 analyses should be interpreted cautiously as these represent a prediction of the functional potential of a community inferred from Illumina amplicon sequencing data. Nevertheless, our communities were well-characterized with regard to placement of ASVs in the per-default reference tree (containing 20,000 full-length 16S rRNA gene sequences), as indicated by the weighted nearest sequenced taxon index (NSTI). We found highly similar NSTI values over all the replicates and treatments ranging from 0.212 to 0.247, which was within the suggested range for well-characterized communities (0.1–0.5[23]). This allowed us to carefully interpret the predicted metagenomes for potential functions and phyla that may react to repeated drought. We did not predict the functional potential of fungal communities due to the high variability of the ITS regions and previous findings indicating that predictions are just slightly better than random for fungal datasets[23].

We further calculated two indices to summarize and quantify the changes in soil microbial community and soil functions. First, we calculated the Bray-Curtis dissimilarity index between drought treatments and control for each analysed microbial community group, as a quantitative indication of treatment differences between microbial communities (Fig. 6, left panels). The graph shows a clear increase in distance from the control with recurrent drought events. Second, we calculated a soil multifunctionality index to quantify the provision of multiple soil processes which are directly linked to ecosystem services[29]. This index is not the expression of a single parameter, but we utilized a wide set of soil parameters (enzymatic activities, soil C and nutrient stock, microbial biomass C, N and P, and root biomass productivity) to assess the response of multiple soil functions to recurrent drought events (Fig. 6, right panel). We show that a single drought event causes a strong decrease in soil functions, while with 10 years of recurrent drought events the index becomes more similar to ambient controls.

## Discussion

The concept of ecological memory is central to ecological theory. It determines an ecosystem's response trajectory after recurrent stress events. Evidence of ecological memory references to a wide range of ecosystems and organisms subjected to climate change-related stress events[30–32]. With drought frequency and severity likely going to increase in many areas of the world[6], knowing how ecosystems will respond to recurrent drought events is crucial. Most studies have focused on short-term effects of drought, while emerging evidence has highlighted the possible induction of legacy effects[15,16,18,33], which are the prerequisite of ecological memory formation. Here we unravel ecological memory formation on microbially mediated soil functions, upon long-term recurrent drought in situ. Specifically, we show the development of biotic legacies, i.e., changes in the microbial community composition and effects on microbially mediated soil functions, with likely repercussions for the whole ecosystem.

We found a large shift in the community composition of bacteria, archaea and fungi, when exposed to 10 years of recurrent drought events, but not for a single drought event. This shift was confirmed by both PLFA analysis and amplicon sequencing of the 16S rDNA and ITS rDNA, at all investigated levels of the community (archaea, bacteria, fungi and AMF). This separation of the community from the control was developed only after multiple recurrent droughts, showing no clear effects up to 5 years of recurrent events (PLFA data only, Supplementary Fig. 5). The formation of an ecological memory relies on either information or material legacies[8,17]. Information legacies are difficult to evaluate in complex communities such as soil microorganisms with no a priori knowledge of the various organismic interactions, and were not analysed. Material legacies can be divided into abiotic and biotic. At our experimental site we did not find persistent changes in edaphic factors (soil pH, soil C and soil

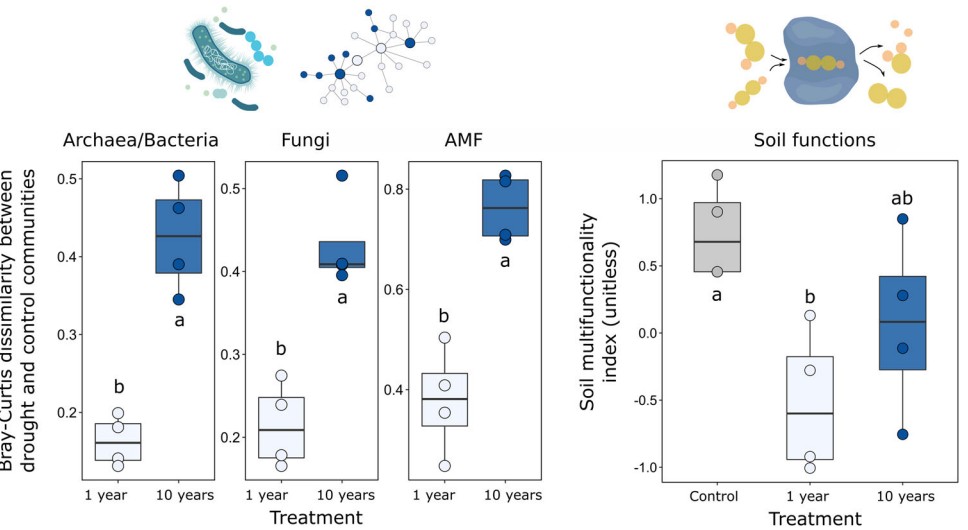

**Fig. 6 Effects of drought on microbial community and soil functions.** Colour indicates soil treatment (control = grey, 1 year = light blue, 10 years = blue). Left panels show boxplots representing the Bray-Curtis dissimilarity index between drought treatments and control for archaea/bacteria ($F = 44.969$, $p = 0.0005$, $n = 4$), fungi ($F = 32.86$, $p = 0.012$, $n = 4$) and AMF ($F = 23.002$, $p = 0.003$, $n = 4$). The right panel shows boxplots representing the soil multifunctionality index (calculated by a wide set of soil parameters including enzymatic activities, soil C and nutrient stock, microbial biomass C, N and P and root biomass productivity) grouped by treatment ($F = 5.514$, $p = 0.027$, $n = 4$). Letters represent results from two-sided Tukey HSD post hoc test. Box centre line represents median, box limits the upper and lower quartiles, whiskers the 1.5x interquartile range, while separated points represents outliers.

nutrient stocks), supporting the notion that it is a biotic legacy of drought that causes divergence from the expected responses to drought[8]. While we did find strong biotic legacies, in that we found strong shifts in the microbial community composition, we cannot completely exclude the influence of abiotic legacies, as there could be other relevant edaphic factors that we did not measure. It is also important to highlight that we did not assess whether the microbial community has reached a new stable state or whether community changes emerged only during the period of drought. Many examples of abrupt transitions to alternative states triggered by climate extremes exist[34], although their implications for ecosystem functioning are not known.

It was recently shown that bacterial community's changes in response to drought are consistent across many studies, indicating that the context of a particular location does not affect the phylogenetic pattern of response[35]. Drought resistance involves deeply conserved traits such as osmolyte production, cell wall features or spore formation[36], which are present in similar groups within soil microbial communities. Indeed, similar to our study, Actinobacteria have previously been found to increase (or show no changes) in relative abundance with drought and Proteobacteria to decline[35,37–40]. Our results were also similar to previous reports on fungi, where Ascomycota were found to increase and Basidiomycota to decrease, indicating that they are likely more sensitive to drought[33,41]. These changes only became evident after 10 years of recurrent drought. We also found that AMF showed a trend in decreasing biomass in response to drought, similar to previous studies[42], but without any change in composition in single drought[42–44], whereas a clear shift was visible after 10 years of recurrent drought. This was especially evident in the fungi of order Diversisporales, which were strongly reduced. This result is particularly relevant given the important role of AMF in system productivity and C cycling[45]. We identified multiple LRGs (Supplementary Fig. 12), indicating phylogenetic groups that responded differently to a subsequent drought, when previously exposed to drought events. All the phyla (or orders) identified as LRGs showed both increased and decreased taxa. The increased taxa represented a large proportion of the total relative abundance, although this should be interpreted

cautiously, as the decrease in relative abundance of some groups inevitably determines the increase of others.

Concomitantly with a biotic legacy of drought via altered soil microbial communities, we observed an ecological memory effect on the potential enzymatic activities, which only emerged after 10 years of recurrent drought events. Indeed, other treatments showed emerging trends, but no real ecological memory formation up to 5 years of recurrent drought events. Similar results were also found for microbial biomass stoichiometric ratios, showing an ecological memory effect. We explain these results via shifts of the microbial community composition in response to drought. After a single drought event we did not find indications of acclimatization of the microbial community structure and function (i.e., a similar composition and strongly altered functions compared to the control). Consequently, the community was unable to cope with the drought stress, which resulted in changes in biomass stoichiometry, as previously shown[46], with no changes in the overall biomass. This is most likely because major pools of C- and N-based microbial metabolites are dynamic in their response to short-time drought, in which MBC was shown to increase and microbial amino acid content to decrease[47]. In contrast, when the microbial community had already repeatedly experienced drought events (i.e., in our 10 years recurrent drought treatment), the composition was significantly altered (representing a biotic legacy) and was better able to cope with drought, as evidenced by the functional capabilities which were more similar to controls than in the single-drought microbial community. We support this with a set of analyses related to community traits. First, we observed an effect of long-term recurrent drought on stoichiometric acquisition strategies. The C:N:P acquisition ratio represents an integral feature to describe soil microbial community demand, linking environmental nutrient availability to the C:N:P stoichiometry of microbial biomass[20–22]. We found that long-term recurrent drought events increased the acquisition strategy towards C- and N-compounds compared to P, while exposure to single drought maintained acquisition strategies similar to control levels. Community-level physiological parameters also revealed a memory effect, where the microbial community re-exposed to long-term drought showed

faster growth and respiration rates and lower CUE values upon rewetting than those subjected to a single drought. This indicates that a shift in microbial community was selected by repeatedly occurring drought stress, allowing microbial communities to re-activate faster upon rewetting of soil.

We also analysed the functional potential of bacterial traits associated to drought resistance. We compared the effects of a single drought with 10 years of recurrent drought to disentangle short-term effects from long-term effects. The community exposed to 10 years of recurrent drought clearly revealed a shift in the bacterial community to being more tolerant to drought, indicated through a range of genes connected to amino acid, carbohydrate and lipid metabolism as well as membrane transporters and other protein families. In general, bacterial cells resist drought stress by decreasing energy consumption, preventing water loss and increasing water retention, and by protecting DNA and preventing protein damage through the accumulation and expression of osmoprotectants[28,48]. We found evidence of multiple genes connected to all these functions being enriched in the bacterial community subjected to 10 years of recurrent drought, which is indicative of a more drought-tolerant bacterial community. Indeed, previous studies have shown that microbial communities that acclimatize to drought usually enhance osmolyte production[49,50], synthesis of capsules and exopolymeric substances to retain water[28,51], dormancy and sporulation[52]. Recent evidence from a long-term drought experiment demonstrated that litter microbial community's physiological response to drought shows a trade-off against growth, by redirecting investment from central metabolism to stress-coping mechanisms[52]. Within the bacterial community, we did not find a shift in the phyla contributing to such functions. Rather, we found that phyla could be connected to both enrichment and depletion at the same time (as previously shown[35]), indicating that the phylum does not represent a high enough level of resolution for identifying bacterial life strategies and responses to stressors[53,54].

The results of this study illustrate several implications for ecosystem responses to drought. To summarize belowground responses to 10 years of recurrent drought as compared to a single drought event, we calculated a soil multifunctionality index which combines the expression of multiple microbial processes and functions[11,55], including activities of 10 extracellular enzymes, soil and microbial biomass nutrients and root biomass productivity. We found that soil multifunctionality is buffered against drought effects when the system is subjected to a history of drought (Fig. 6). Given the pivotal importance of soil microbial communities for multiple ecosystem functions and services[56], our results can have several implications. For example, it was shown that plant fitness is strongly linked to the rapid responses of soil microbial community structure to drought[57], and that changes in the relative abundance of specific bacterial taxa are associated with increased drought tolerance of plants[58]. Moreover, the contribution of heterotrophic $CO_2$ emissions from soil can become dominant during periods of drought followed by rewetting pulses[59] and can account for up to 60% of net ecosystem $CO_2$ release during a dry season[60]. As recently proposed, incorporating microbial community composition into ecosystem process models will help in predicting ecosystem responses to disturbances. This may be especially true for disturbances or other deviations from steady-state conditions, such as drought events[61]. Increasing climate variability[6] will certainly add a further layer of complexity in predicting ecosystem response to disturbances and should be considered more explicitly in future studies.

The biotic legacy that we found at the microbial community level in our study, may be also connected to biotic legacies within the plant community. Indeed, a strong link exists at the root interface between plant and soil microbial community[62]. This is, for example, supported by the strong changes we observed in the arbuscular mycorrhizae community, which depends to a large extent on plants. At our experimental site, the plant community was indeed altered in response to recurrent droughts (Bahn et al. 2020, unpublished information). However, the influence between plant and microbes is bi-directional, where plants can shape the soil community and its drought response[63] and in turn the soil microbes can have important consequences for the plant diversity, community composition and survival[64,65]. Therefore, it is not possible to conclude to which extent legacy effects in the plant community could determine legacies in soil microbes and vice versa. Drought can also modify plant C inputs into soil. For example, drought can alter the quantity and quality of root exudates[66,67], and legacy effects have been shown to cause a reduction of belowground C inputs[16]. Plant litter chemistry might also be affected by direct drought effects and via changes in plant community caused by long-term drought[68], which in turn might alter soil microbial communities[69,70]. A recent study showed that plant litter chemistry constrains microbial community gene expression and metabolite production associated with drought stress tolerance[52]. Although plant C input was not the focus of this experiment, plant community dynamics (and associated belowground C inputs) are thus likely an important player in the response of microbial communities to drought.

As drought events become more frequent, the response of an ecosystem to each new disturbance will likely be contingent on the history of precedent extreme events. Temporal phenomena are fundamental to ecology, as the enduring influences of past experiences and changing conditions often unfold over time[71,72]. In this context, ecological memory, i.e., 'the capacity of past states or experiences to influence present or future responses of a community'[2], is critical to predict a disturbance effect on ecosystems. The memory effect is transmitted as legacies of species and materials, which are prerequisites. We here examined the extent to which a soil microbial community subjected to long-term recurrent drought diverges from the response to a single drought event. Via a tailored experimental setting and repeated sampling campaigns, we were able to disentangle the effects of single drought vs recurrent drought events and assess whether ecological memory formation, induced by recurrent drought events, expresses during a subsequent drought. We show two key findings: (i) recurrent drought strengthens shifts in the composition of both bacterial and fungal communities; (ii) soil multifunctionality is buffered in response to drought by previous recurrent drought events. We further suggest that microbial community shifts and changes in the microbially mediated cycling of carbon and nutrients likely reflect community-dependent strategies in response to drought and likely controlling soil functions.

This study demonstrates how reinforcing interactions between consecutives stress events may affect the response trajectory of ecosystems to climate change. Such information is of utmost importance for understanding climate change feedbacks. Specifically, long-term exposure to recurrent drought events can select for a more tolerant soil microbial community and create a memory effect, modifying soil processes that mediate and enhance the resistance of soil functions to drought.

## Methods

**Site description.** The study site is located in the Austrian Central Alps near Neustift, Stubai valley (47°07′45″N, 11°18′20″E; Supplementary Fig. 15) at 1850 m a.s.l. It is part of a mountain meadow that is annually cut for hay production at peak biomass (late July/early August) and fertilized every 3–4 years. The soil is a Dystric Cambisol (Food and Agriculture Organization-soil classification system) with a pH of 5.4 (in CaCl₂). The vegetation is dominated by highly productive perennial grasses and herbs including *Anthoxanthum odoratum*, *Festuca rubra*,

*Alchemilla vulgaris*, *Leontodon hispidus* and *Trifolium repens* and the meadow is generally characterized by a comparatively high primary productivity and high soil $CO_2$ efflux rates[73–75]. The mean annual temperature and the mean annual precipitation are 3 °C and 1100 mm, respectively[19,73–76].

**Sampling campaigns and experimental set-up.** Samples were collected from a unique drought experiment in a mountain grassland that was initiated in 2008. From 2008 to 2017 drought was simulated annually by equipping plots with rainout shelters (3 × 2 m, $n = 4$) for 8–10 consecutive weeks during the growing season and excluding approximately 34% of the annual precipitation[76]. In addition, in 2009, 2011 and 2017 new rainout shelters were installed to compare drought responses of plots subjected to a different number of drought events. Hence, in 2011 samples were collected from plots subjected to 1, 3 and 4 drought events, in 2012 from 2 and 5 drought events, and in 2017 for comparing 1 and 10 drought events (Supplementary Fig. 15). In 2017, drought plots were equipped with rainout shelters from May 17th until August 1st. During the drought period, the soil water content (SWC) in the 10-year and 1-year treatments ranged around 10% (volumetric soil water content at 10 cm depth), whereas it remained in the range of 25–50% in control plots (Supplementary Fig. 1). Drought was terminated by simulating a heavy rainfall event (20 mm of previously collected rainwater was applied to each of the plots) upon removal of the rainout shelters followed by subsequent exposure to natural precipitation. For each plot ($n = 4$ for each treatment) three replicate soil cores (2 cm diameter) were collected from a depth of 0–10 cm. Soil samples were collected at the end of July (after around 10 weeks of drought) for each campaign and transported to the laboratory (University of Vienna) where they were carefully sieved with a 2 mm sieve and roots were manually removed. Subsamples of soil were immediately frozen for further analysis of the soil microbial community through PLFA (in the 2011 campaign, only control and 1-year drought were analysed for PLFA). The remaining soil was immediately used to measure water content, potential enzymatic activity and for extractions of available nutrients, carbon and microbial biomass. In 2017, we expanded the set of analyses carried out. First, more enzymes were tested (Supplementary Table 3) and microbial communities were investigated by both PLFA and amplicon sequencing. Root biomass production during the 2017 campaign was also measured through in-growth cores. Cores were placed in the soil from May 31st to July 31st and root growth was measured over a period of 62 days. This data was only used to generate a soil multifunctionality index as it is part of another manuscript (Bahn et al., in preparation).

**Analysis of soil microbial growth, respiration and carbon use efficiency.** Soils were subjected to a laboratory incubation (24 h), where we applied $^{18}O$-labelled water to simulate a short time rewetting event and simultaneously study changes in microbial growth, respiration and CUE upon rewetting. We measured soil microbial activity (mass-specific gross growth rates, mass-specific respiration and CUE) by using $^{18}O$-$H_2O$ incorporation into microbial DNA[77]. Briefly, approximately 400 mg of fresh soil was incubated for 24 h at temperatures resembling the ambient field conditions during sampling (ca. 20 °C) after labelling them with $^{18}O$-$H_2O$ to reach an overall enrichment of 20 at% (97.0 at% water, Campro Scientific). Duplicates, which received the same volume of molecular-grade non-labelled water served as natural abundance controls. In order to calculate microbial respiration, gas samples were taken at the start and end of the incubation period and subsequently analysed for respective $CO_2$ concentrations using a Trace GC Ultra (ThermoFischer, Waltham, USA). Following the collection of last gas sample, incubated soil samples were frozen in liquid nitrogen and stored at −80 °C. Consecutively, total microbial DNA was extracted via a DNA-extraction kit (FastDNA™ SPIN Kit for Soil, MP Biomedicals Santa Ana, USA) and quantified by PicoGreen assay (Quant-iT™ PicoGreen® dsDNA Assay Kit; ThermoFischer, Waltham, USA). The abundance of $^{18}O$ in the extracted DNA was determined for labelled and control samples via a thermochemical elemental analyzer (TC/EA ThermoFisher) coupled via a Conflo III open split system (ThermoFisher) to an Isotope Ratio Mass Spectrometer (Delta V Advantage, ThermoFisher). The synthesis of new DNA was calculated as the difference in $^{18}O$ abundance between the labelled and non-labelled samples ($O_e$ as the $^{18}O$ at% excess of the labelled sample), the $^{18}O$ enrichment ($O_l$ in at%) and the factor of 31.21 describing the proportional mass of O (%) in an average DNA molecule.

$$DNA_{produced} = O_t * \frac{O_e}{100} * \frac{100}{O_l} * \frac{100}{31.21}$$

The microbial growth rate (the amount of produced microbial C) [ng C g$^{-1}$ DW h$^{-1}$] was calculated by using a conversion factor (fDNA = $C_{mic}$/DNA$_{mic}$), which describes the relation between microbial DNA and microbial C, and the amount of newly produced DNA (for details see[77]).

$$C_{produced} = \left( \frac{C_{mic}}{DNA_{mic}} \right) * DNA_{produced}$$

We expressed the amount of carbon that was taken up by the microbial community [ng C g$^{-1}$ DW h$^{-1}$] as the sum of carbon allocated to respiration plus biomass production (growth).

$$C_{uptake} = C_{respiration} + C_{growth}$$

The soil microbial CUE was calculated as the ratio of the C that was invested into growth to the total C that was taken up[78].

$$CUE = \frac{C_{growth}}{C_{uptake}}$$

We expressed mass-specific respiration rates [mg $CO_2$-C g$^{-1}$ $C_{mic}$ h$^{-1}$] as ratio between respiration-derived C and MBC over time, whereas the mass-specific growth rate [ng C ng $C_{mic}^{-1}$ d$^{-1}$] was calculated as the ratio of $C_{growth}$ over $C_{mic}$.

$$\text{Mass specific respiration rates} = \frac{C_{respiration}}{C_{mic}} / time$$

$$\text{Mass specific growth rates} = \frac{C_{growth}}{C_{mic}} / time$$

**Analysis of soil, microbial stoichiometry and available nutrients.** All soil samples were analysed for gravimetric water content, microbial biomass C, N and P, extractable organic C (EOC), extractable organic nitrogen (EON), nitrate ($NO_3^-$), ammonium ($NH_4^+$), total free amino acids (TFAA), extractable organic P (EOP) and extractable inorganic P (EIP). Soil aliquots were dried at 60 °C for 72 h, weighed and finely ground for subsequent analysis of bulk total C and N contents by EA-IRMS (EA 1110; CE Instruments, Milan, Italy), coupled to a Finnigan MAT Delta Plus IRMS (ThermoFisher Scientific, Waltham, MA, USA). Soil P was measured photometrically based on phosphomolybdate blue reaction[79] with prior ignition at 450 °C for 5 h in order to convert organic P to inorganic forms[80]. Microbial biomass C, N and P (MBC, MBN and MBP) were measured following the chloroform fumigation-extraction technique[81], with some modifications. Briefly, 4 g of soil from each treatment was weighed in two different containers. To the first set of containers, 30 mL of 1 M KCl solution (for C and N) or 0.5 M NaHCO$_3$ (for P) was added and shaken for 1 h, before filtering through ash-free cellulose filters and extracts kept frozen at −20 °C for later analyses. The other half of samples was fumigated in a desiccator with chloroform for 48 h and then extracted as the non-fumigated samples. Total C and N content of the KCl extracts was analysed by a TOC/TN analyzer (TOCVCPH E200V/TNM-122V; Shimadzu, Vienna, Austria). The difference between fumigated and non-fumigated samples was divided by 0.45[81] and considered as the MBC, while a factor of 0.54[82] was used to calculate MBN. Inorganic phosphate was measured photometrically based on phosphomolybdate blue reaction[79] with or without prior alkaline persulfate oxidation for measuring total dissolved P and inorganic P, respectively[83]. MBP was calculated as the difference between the fumigated and non-fumigated samples. Ammonium and nitrate were determined with colorimetric methods, $NH_4^+$ with a modified indophenol reaction[84] and $NO_3^-$ with the VCl$_3$/Griess assay[85]. EON was calculated by subtracting from N values of non-fumigated samples the sum of $NH_4^+$ and $NO_3^-$. Amino acid concentrations were quantified by a modified fluorometric OPAME procedure based on[86], optimized for free amino acid measurement in protein hydrolysates[87]. Inorganic N, TFAA and P assays were run in a microtiter plate format (μQuant Qx200, Bio-Tek Instruments, Bad Friedrichshall, Germany).

**Potential extracellular enzymatic activity.** The potential extracellular enzymatic activity (EEA) of 10 different enzymes related to C-, N-, P- and S-cycle was measured (Supplementary Table 3) through a photometric method[88], with some modifications. Briefly, 1 g of sieved soil was suspended in 100 mL of sodium acetate buffer (50 mM, pH 5) and ultrasonicated at low energy (total energy output of 0.35 kJ). Then, 200 μL of soil suspension and 50 μL of substrate were pipetted into black microtiter plates in six analytical replicates. Aminomethylcoumarin (AMC) was used for calibration of the proteases (Supplementary Table 3), while methyl-lumbelliferyl (MUF) was used for calibration of all the remaining enzymes. Plates were incubated in the dark at 20 °C and fluorescence was measured every 30 min for six times at an excitation wavelength of 365 nm and an emission wavelength of 450 nm (Tecan Infinite M200 fluorimeter, Werfen, Austria). Activity was then calculated as the increase in fluorescence over time.

**PLFA analysis.** Soil microbial biomass and microbial community composition were estimated by extracting PLFAs from freeze-dried soil samples using the same procedure as described in[89] with some modifications. Total lipids were extracted from soil using a chloroform/methanol/citric acid buffer and fractionated by solid-phase extraction. The NLFA fraction was collected by eluting samples with chloroform while the PLFA fraction was collected by eluting samples with methanol. After an internal standard (19:0) was added, NLFAs and PLFAs were converted to fatty acid methyl esters (FAMEs) by transesterification. Samples were analysed for identification and quantification using a GC (7890B GC System; Agilent, Santa Clara, CA, USA) connected to a TOF/MS (Pegasus HT; LECO Corporation, Saint-Joseph, MI, USA). Samples were injected in splitless mode (injector temperature 220 °C) and separated on a DB5 column (60 m × 0.25 mm × 0.25 μm; Agilent, Vienna, Austria) with 1.5 mL min$^{-1}$ He as the carrier gas (GC

program: 1 min at 80 °C, 30 °C min$^{-1}$ until 150 °C, 1 min at 150 °C, 2 °C min$^{-1}$ until 200 °C, 4 °C min$^{-1}$ until 230 °C, 15 min at 230 °C, 30 °C min$^{-1}$ until 290 °C and 5 min at 290 °C). FAMEs were identified using mixtures of bacterial and fungal FAMEs (Bacterial Acid Methyl Ester CP Mixture (Mtreya LLC, State College, PA, USA) and 37 Component FAME Mix (Supelco, Bellefonte, PA, USA)) and evaluated against NIST MS library for confirmative identification. FAMEs were quantified against the internal standard (19:0). We used the markers 18:1ω9, 18:2ω6,9[88] and 16:1ω5 for total fungi, although 16:1ω5 is a marker used for AMF it can also be present in bacteria[90]. Therefore, the NLFA 16:1ω5 was also measured, as this represents a more specific marker for AMF[91]. The sum of i15:0, a15:0, i16:0 a16:0, i17:0, a17:0 and a18:0 was used as general Gram-positive bacterial marker and 16:1ω7, cy17:0 and cy19:0 for Gram-negative bacteria[92]. The marker 10Me16:0 and 10Me17:0 were used for Actinobacteria[92] and summed with general Gram-positive, Gram-negative and 14:0 and 15:0 to give total bacterial PLFAs[93]. For a number of minor peaks not present in the external standard, the double bond position could not be identified with confidence and therefore they were assigned to the general group altogether with the markers 16:0, 18:0, 17:1ω10 and 19:1ω9.

**Amplicon sequencing.** DNA was extracted from freeze-dried samples using the FastDNA™ SPIN Kit for Soil (MP Biomedicals Santa Ana, USA). Extracted DNA was quantified (Quant-iT™ PicoGreen® dsDNA Assay Kit; ThermoFisher, Waltham, USA) and manually normalized to 5 ng µL$^{-1}$ for each individual sample. Normalized DNA was used to generate amplicon libraries of archaeal, bacterial and fungal communities using a two-step barcoding approach[94]. In the first PCR step, target amplicons were generated with primers modified at the 5′ end to include a 16 bp head sequence [5′-GCTATGCGCGAGCTGC-3′]. In the second PCR step, products from the first reaction were amplified with barcoding primers that targeted the 16 bp head sequence and included a library-specific 8 bp barcode at the 5′ end. Archaeal and bacterial amplicons were generated using primers 515F[95] and 806R[96] that targeted the V4 region of the 16S rRNA genes. Fungal amplicons were generated for internal transcribed spacer region 1 (ITS1) using primers ITS1F and ITS2[97,98], additionally modified by[99] and region 2 (ITS2) using primers gITS7ngs and ITS4ngs[100]. In-house mock communities for both SSU rRNA genes and ITS rDNA, and extraction blanks and water control were included for quality control of sequencing data. In addition, a nested PCR approach was used to target AMF. In the nested approach, a sequence of >2 kbp spanning the whole ITS rDNA region was amplified using a set of partially degenerate primers SSUmAf1-2 and LSUmAr1-4[101]. The resulting PCR product was then used as a template to amplify ITS2 using the 5′-modified primers 5.8S[86] and ITS4[97,98]. The complete overview of primers used in this study is given in Supplementary Table 4.

All oligonucleotides were obtained from Biomers.net (Ulm, Germany). First-step PCRs were performed in three analytical replicates of each sample (2 µL DNA, 5 ng µL$^{-1}$) in a total reaction volume of 25 µL containing 1x Dream Taq Green Buffer including 2 mM MgCl$_2$ (ThermoFisher Scientific), 0.2 mM dNTP (ThermoFisher Scientific), 0.1 mg mL$^{-1}$ bovine serum albumin (ThermoFisher Scientific), 0.25 µM of each forward and reverse primer and 0.025 U Dream Taq DNA Polymerase (ThermoFisher Scientific). Amplification conditions for SSU rRNA-targeted PCRs (primers 515 F–806 R): 94 °C for 4 min, 25 cycles of 94 °C for 30 s, 52 °C for 30 s and 72 °C for 45 s, followed by final elongation at 72 °C for 10 min. Amplification conditions for ITS1-targeted PCRs (primers ITS1F–ITS2): 94 °C for 4 min, 25 cycles of 94 °C for 30 s, 52 °C for 30 s and 72 °C for 45 s, followed by final elongation at 72 °C for 10 min. Amplification conditions for ITS2-targeted PCRs (primers gITS7ngs–ITS4ngs): 94 °C for 4 min, 30 cycles of 94 °C for 30 s, 52 °C for 45 s, and 72 °C for 60 s, followed by final elongation at 72 °C for 10 min. Amplification conditions for SSU rRNA–ITS–LSU rRNA-targeted PCRs (primers SSUmAf1-2–LSUmAr1-4): 98 °C for 3 min, 30 cycles of 98 °C for 10 s, 58 °C for 30 s, and 72 °C for 3 min, followed by final elongation at 72 °C for 10 min. PCRs were performed in three analytical replicates of each sample (2 µL DNA, 5 ng µL$^{-1}$) in a total reaction volume of 25 µL per sample containing 1x Phusion HF Buffer containing 1.5 mM MgCl$_2$ (ThermoFisher Scientific), 0.2 mM dNTP (ThermoFisher Scientific), 1 µM of each forward and reverse primer, and 0.02 U Phusion DNA Polymerase (ThermoFisher Scientific).

Amplification conditions for AMF-ITS2-targeted PCRs (primers AM5.8S_ILfor–ITS4): 94 °C for 4 min, 25 cycles of 94 °C for 30 s, 52 °C for 45 s, and 72 °C for 60 s, followed by final elongation at 72 °C for 10 min. The first-step PCR was screened by agarose gel electrophoresis, pooled, purified using the SequalPrep™ Normalization Plate Kit (ThermoFisher Scientific) and used for second-step PCRs. Second-step PCRs were carried out in 50 µL reactions consisting of same concentration of reagents as the first-step PCRs while using 10 µL of template and 0.8 µM barcoding primers (5′-BARCODE-HEAD-3′). PCR products were screened by agarose gel electrophoresis and purified and normalized using the SequalPrep™ Normalization Plate Kit resulting in an equimolar library. Samples were pooled and concentrated (innuPREP PCRpure PCR Cleanup Kit, analytikjena, Germany). The final pooled library was sent to Microsynth AG (Balgach, Switzerland) for sequencing on a MiSeq system (Illumina) using the TruSeq Nano DNA Library Prep Kit (Illumina, Cat FC-121-4001). The MiSeq was run in the 2 × 300 cycle configuration using the MiSeq Reagent kit v3 (Illumina, Cat MS-102-3003). Resulting datasets were deposited in the NCBI Sequence Read Archive under BioProject accession number PRJNA694357.

Paired reads were demultiplexed according to[94] and were processed into merged ASVs using DADA2 v. 1.10.0[102] with standard pooled processing (pool = TRUE) and with default parameters. ASVs predicted for fungi (ITS1, ITS2 and AMF-ITS2) were screened with[103] ITSx v. 1.0.11. All ASVs were screened for chimeras using UCHIME[104]. ASVs were taxonomically classified using the RDPclassifier[105] as implemented in Mothur[106] v 1.39.5. 16S-V4 regions were classified using[107] the SILVA SSU database 132 and ITS regions were classified using the Warcup training set V2[108].

**Calculations and statistical analysis.** All analyses were performed in R V3.6.0[109]. Following amplicon sequencing, data handling and manipulation were undertaken using the phyloseq package[110]. Data was filtered by removing sequences matching 'Mitochondria', 'Chloroplast' and 'Eukaryota' from the archaeal and bacterial community, and by removing 'Glomeromycota' from the fungal community as the analysis of Glomeromycota was carried out with a specific primer. Also, in order to minimize the inflation of rare species in the community analysis, we removed ASVs that were not present more than two times in at least 20% of the samples. This filtered ASV matrix was used for all downstream analyses concerning amplicon sequencing data. For α-diversity analysis, all samples were subsampled (rarefied) to the minimum sample size using bootstrap subsampling at 1000 iterations to account for library size differences (5809 reads per sample for SSU V4, 9959 reads per sample for ITS2, 2270 reads per sample for ITS1, 2474 reads per sample for AMF-ITS2). α-Diversity indices were generated with the function 'plot_richness' from the package phyloseq. For β-diversity analysis, library size normalization was carried out using the geometric mean of pairwise ratios[111]. Detection of differentially abundant ASVs was carried out using the DESeq2 package[112]. Plots were generated using ggplot2 package[113]. We classified microbial ASVs into LRGs similarly to Meisner et al.[33], but with some modifications. These are defined as groups of organisms with similar response to a change in their environment. Here, we defined groups of ASVs that differ between 10-year and 1-year treatments, which therefore identify groups of microorganisms that do not simply response to drought but that are specific to long-term drought effects. LRGs were extracted from the filtered ASV tables, as described in the 'Methods' section, in order to minimize the inflation of rare species in the community analysis. Detection of differentially abundant ASVs was carried out using the DESeq2 package[112]. LRGs were grouped for those ASVs showing either an enrichment in the 10-year treatment (positive) or a decrease (negative). The ASVs were then phylogenetically assigned to the corresponding phylum (or order for the AMF dataset) and expressed as relative abundance of the 10-year microbial community.

For all individual variables collected in 2017, we used a one-way ANOVA to test the main effects of treatments (control vs 1 year vs 10 years). We checked for homogeneity of variances and normality of residuals by inspecting the plot of standardized residuals vs predicted values, frequency histogram and QQ-plot[114] (reported in Supplementary Statistical Report). Given the high number of tests presented, we corrected the p-values presented for false discovery rate (FDR), using the Benjamini-Hochberg (BH) procedure with the R function 'p.adjust()'. We present the ANOVA results with and without correction in Supplementary Table 5. A post hoc test (Tukey HSD test) was used to compare differences between individual treatments. The package 'effectsize'[115] was used to calculate eta-squared effect size ($\eta^2$) and 90% confidence intervals. Data integrating previous campaigns (2011 and 2012) was analysed by linear models using the function 'lm'. Data from potential enzymatic activity, PLFA and amplicon sequencing was also analysed by means of permutational analysis of variance (9999 permutations) using the 'adonis' function in the package vegan[116]. Results from the permutational multivariate analyses were visualized by means of non-metric multidimensional scaling (NMDS). For this, we used the function 'metaMDS' from the package vegan.

To compare quantitative differences between control treatments and different years of drought subjection, we calculated the Bray-Curtis dissimilarity index with the function 'vegdist' of the package vegan. The index was calculated as an averaged pairwise comparison between plots subjected to drought and the control treatment. We calculated this index in the 2017 campaign from the amplicon sequencing data, for the 16S, ITS1 and AMF primer. We also calculated a response ratio, as the natural logarithm of drought treatments divided by the average of the control (Ln (RR)) for the sampling campaigns of 2011 and 2012. The response ratio is commonly used in ecological studies as a measure of experimental effect because it quantifies the proportionate change that results from an experimental manipulation[117]. Moreover, it allowed us to compare different variables (enzyme activities and acquisition strategies) in the 2011 and 2012 campaigns, and to relate changes within the control to previous years of drought. We calculated stoichiometric ratios of microbial acquisition strategies by dividing enzymes related to C-cycle (BG, CB, LIP and XYL) into either N-cycle-related enzymes (END, NAG, LAP, TAP) or P-cycle-related enzyme (PHOS) and also the ratio of N-cycle-related enzymes to P-cycle-related enzymes. For the 2011 and 2012 campaigns we used a reduced set as specified in Supplementary Table 3.

We also calculated a multifunctionality index, which represents the expression of multiple ecosystem processes and services together[29]. We calculated this index using only soil-related parameters including soil nutrients (soil C, N and P), microbial stoichiometry (biomass C, N and P), plant productivity (belowground net primary productivity) and microbially mediated organic matter decomposition (potential enzymatic activities). Thus, representing a soil multifunctionality index.

In order to obtain a computable index for each treatment, we standardized each of the variables measured using the *z*-score transformation. These standardized soil-function values were then averaged to generate a multifunctionality index[55].

We used the filtered ASV matrix to predict the soil bacterial metagenomes using PICRUSt2[23] with implemented tools HMMER (http://hmmer.org/), EPA-ng[118], GAPPA[119] and CASTOR[120]. The ASV file was converted into a.biom file using the R package biomformat[121] and PICRUSt2 was executed to obtain the stratified output based on per-sequence contributions. We downloaded the KEGG Orthology (KO) from https://www.genome.jp/kegg-bin/get_htext and created a custom file that allowed to map KO numbers to broader categories as provided by the PICRUSt2 tool. The weighted nearest sequenced taxon index (NSTI) was found to be highly similar over all the replicates and treatments ranging from 0.212 to 0.247 and indicated a well-characterized community with regard to the placement of ASVs in the reference tree. This allowed us to carefully interpret the predicted metagenomes for potential functions and phyla that may react to repeated drought. We applied DESeq2 to identify the predicted functions that were differentially abundant among treatments including the phyla predicted to be contributing to these functions.

**Reporting summary**. Further information on research design is available in the Nature Research Reporting Summary linked to this article.

## Data availability

16S-V4 regions were classified using the SILVA SSU database 132 (https://www.arb-silva.de/). The data generated in this study (excluding data generated via the amplicon sequencing analysis) is provided in the Supplementary Information (Supplementary Data 1). The overall data generated in this study (including data generated via the amplicon sequencing analysis) has been deposited in the Zenodo repository database under accession code DOI: 10.5281/zenodo.5117955. The sequencing data generated in this study have been deposited in the NCBI Sequence Read Archive under BioProject accession number PRJNA694357.

## Code availability

The R code supporting the findings presented here is available from the corresponding authors on request.

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

## Acknowledgements

We thank Margarete Watzka and Ludwig Seidl for assistance in the laboratory. We further thank Jan Jansa for guidance with AMF-targeted amplicon sequencing. The long-term drought treatment was maintained by projects funded by the Austrian Science Fund (FWF, project number P22214-B17 and I 1056) and the Austrian Academy of Sciences (ESS-programme, project ClimLUC). Open access funding provided by University of Vienna.

## Author contributions

M.B. designed the experiment. M.B., A.R., L.F. and R.H. carried out the field campaigns, and A.C., H.S., L.F., D.Z., V.M., M.J. and P.G. the laboratory analyses. H.S. and C.W.H. processed amplicon sequencing data which was analysed by A.C. and H.S.; A.C. analysed the remaining data. A.C., A.R., H.S. and L.F. wrote the original draft, and all authors reviewed the manuscript.

## Competing interests

The authors declare no competing interests.
