## [Peer Review File. · Nature Communications]

Ecological memory of recurrent drought modifies soil processes via changes in soil microbial communityREVIEWER COMMENTS

Reviewer #1 (Remarks to the Author):

The authors studied drought effects on microbial community and their function. They compared a long-term drought (10 years) with a single drought event and ambient conditions to show ecological memory formation of microbially mediated soil processes in response to drought. They applied potential extracellular enzyme assays, PLFA analysis, amplicon sequencing and quantitative physiological measurements such as growth rate and carbon use efficiency to soil samples from the field experiment. They find that recurrent droughts shift the microbial community composition and their putative functions compared to ambient and single drought events. Based on the results they conclude that an ecological memory of drought in soil processes is caused by a shift in microbial community composition, which buffers soil functionality against drought effects.

The results are novel because they compare field experiments with a long-term recurrent drought event to a single drought event. Although the methods used in the paper are standard and routine, such data from a unique experimental combination based on period of drought treatment are rare and therefore of interest to the community. Overall, I enjoyed reading the paper. I have some minor comments aimed at improving the paper.

Conclusions are original, novel and well supported by the generated data. Functional predictions made using PICRUSt are weak but they fit well with the other more quantitative data. The functional genes identified are also on expected lines with no major surprises. The authors don't attribute the changes in community structure and function as well as other physiological indicators such as stoichiometry on changes in plant community and litter chemistry and root exudation which are highly likely after 10 years of drought. These only get a passing mention at the end of the discussion section.

Work is convincing with plenty of data of different kinds supporting the overall story. The discussion section is very well written with relevant literature properly cited and discussed in the context of the study. Some more papers directly relevant to the discussion on drought adaptation of bacterial communities could be useful. There are clearly many soil relevant papers on drought and osmoprotection that are missing, which could enhance the discussion thus making it more credible. Instead of listing the papers here, I recommend the authors go through my recently published paper on microbial gene expression and metabolite response to a legacy of long-term drought: Malik, A.A., Swenson, T., Weihe, C. et al. Drought and plant litter chemistry alter microbial gene expression and metabolite production. *ISME J* 14, 2236–2247 (2020). <https://doi.org/10.1038/s41396-020-0683-6> Here we also discuss the legacy effects of drought in plant community that changes the chemistry of plant organic matter which are resources for microbes.

Around line 433, you mention "After a single drought event... the community was unable to cope with the drought stress". This is supported by changes in biomass stoichiometry, but I wonder why it does not reduce microbial biomass, in fact there is a trend in the opposite direction.

Information on incubation for growth rate and CUE measurements are not provided in as much detail as other methods. These are instead provided in supplementary section. The results of these analyses are also not presented in the main text, although these are quite valuable and add to the story.

Line 334: not clear what is meant by "high functional complementary within bacterial phyla". The authors could clarify it a bit.

Line 434: should be "change in biomass stoichiometry" – typo

460: "We found evidence of multiple genes connected to all these functions being enriched in the bacterial community subjected to 10 years of recurrent drought. Consequently, being indicative of a drought adapted bacterial community." Sounds like one whole sentence instead of two.

Some of the paragraphs are very long. Consider splitting them based on sub-themes.

Statistical analyses are appropriate and provided in sufficient detail. All data are either provided in the supplementary section or made publicly available in repositories.

All the best
Ashish Malik

Reviewer #2 (Remarks to the Author):

Canarini et al., investigated the effects of recurrent drought events and single drought events on microbial community and function. Their results showed a tendency of ecological memory of microbially mediated processes to a history of drought events. The authors further concluded that the ecological memory of drought in soil processes is caused by a shift in microbial community composition. However, I had some difficulties in understanding the causal relationship here. I am afraid that the author may over-interpret their results based on their experimental design and their data records. This is my major concern. In addition, I found several typos and grammar errors. I would suggest the authors double-check their writing before their resubmission.

Line 38-40. I am afraid of the over-interpretation here.

Line 104 -119. I recognize the valuable contribution of long-term experiments. But, did you consider the climate and vegetation dynamics over the long term?

Fig. 1. What does dm in γ -lab stand for? Why not normalizing per gram soil.

Fig. 3. How did you classify enzymes related to the CNP cycle? Some enzymes are related to at least both of them.

Fig. 5. Why not showing the values from control in the left panels?

Line 515. I did not find related information in Supp Fig. 1.

Line 516-517. How did you evaluate "grazed for a few days", if considering the substantial grazing effects on soil microbial processes?

Line 517. Will it be better to list the full name for FAO?

Line 518-520. The dominant species are required.

Line 520-521. In which year range?

Line 527. To make your study more comparable with others, you may calculate how much precipitation intercepted by rainout shelters?

Line 529. Be consistent with "rain-out" or "rainout" across your MS.

Line 676. For ANOVA and some other data analysis, data normality and residual distribution should be reported.

Reviewer #3 (Remarks to the Author):

Remarks to the authors:

Manuscript summary:

The rationale for this study is understanding the impacts of drought events (whose frequency is associated with climate change) and its effect on ecological processes in soils. The authors test the hypothesis that drought events can lead to "ecological memory formation", which is defined as the capacity of past states to influence present or future responses. The focus of this study is the impacts of abiotic (edaphic) and biotic (microbial) factors into this ecological memory formation. Specifically, this effort focuses on a very unique experimental field site where ten years of simulated recurrent drought is compared to a single drought event (as well as other lengths of simulated drought events). The main finding of this study is that the ten year recurring drought response of the soil is different from a single drought event, which the authors posit can provide stability for soil functionality against drought effects. Further, this impact is mainly biological and

not observed in abiotic measurements. The authors conclude that the shift in microbial community composition indicates an ecological memory of drought in soil processes.

This paper was well-written, and the introduction was helpful in setting the rationale for this study. In a brief literature search of drought microbial legacy effects, I found it justified that the impacts of drought on shifting microbial communities were well-studied and that this study was novel for its focus on specifically long-term recurrent drought events (> 1 year).

The focal results in this effort in my opinion were the lack of differences observed in soil edaphic properties and the significant differences observed in both community structure and the enzyme potential. The observation of a shift between one year and ten year drought events were clear to me. My challenge in justifying the findings can be identified in the key discussion paragraph on Lines 492 and 503.

"Via a unique experimental setting and repeated sampling campaigns, we were able to disentangle effects of single drought vs recurrent drought events and assess ecological memory formation. We show two key findings: (i) recurrent drought strengthens shifts in the composition of both bacterial and fungal communities; (ii) soil multifunctionality is partially restored by recurrent drought events. We further suggest that microbial community shifts and changes in the microbially mediated cycling of carbon and nutrients, likely reflects community-dependent strategies to adapt to drought and controlling soil functions."

I do not find the two key findings adequately justified. Mainly, I agree with the authors that the shift is present but the formation of "legacy" or "memory" is not clear to me. I do not see results that STRENGTHEN THE SHIFT in composition in the ten year samples. Figure 1 shows that the ten year is more similar to controls via microbial biomass but does not directly show a change in the magnitude of the shift, only that 1-year and 10-year are different. Figure 2 results also show differences between control, 1 year, and 10 year generally but do not in my opinion provide insights into the magnitude of this shift. And similar thoughts on the other figures. The authors bring in some other datasets in the Supplemental to include other drought events. Firstly, I found it difficult to synthesize this extra dataset within the context of the main paper, and feel like it has an opportunity to maybe address my concerns (if data can be obtained from these samples) but is not currently used to address the identification and quantification of the "legacy effect".

Second, I am not convinced that soil functionality is restored and that the current data informs a "restoration". Partially, this is due to the lack of clarity on what the legacy effect is, and partially I am not clear on if restoration to a "baseline" condition was part of this experiment. One can argue that soil "multifunctionality" is shown to return to somewhat baseline in Figure 3 but the text is not clear to me on what results justify this statement. Perhaps it is the identification of the inferred functions from PICRUST that are the justification. As the authors point out themselves, I think this should be done with extreme caution. If this is a key finding, I believe that functional assays be developed based on the hypotheses that can be developed from these findings to justify the conclusion. In a review of literature, I came across this effort (<https://www.frontiersin.org/articles/10.3389/fmicb.2018.00294/full>), which defines legacy response groups and suggest that it may provide an approach to authors if they find it helpful. There are likely other model strategies in other ecosystems besides soils to quantify the "legacy" effect, but I think this would help strengthen the findings beyond a shift to actual "memory".

RESPONSE TO REVIEWER COMMENTS

Reviewer #1 (Remarks to the Author):

The authors studied drought effects on microbial community and their function. They compared a long-term drought (10 years) with a single drought event and ambient conditions to show ecological memory formation of microbially mediated soil processes in response to drought. They applied potential extracellular enzyme assays, PLFA analysis, amplicon sequencing and quantitative physiological measurements such as growth rate and carbon use efficiency to soil samples from the field experiment. They find that recurrent droughts shift the microbial community composition and their putative functions compared to ambient and single drought events. Based on the results they conclude that an ecological memory of drought in soil processes is caused by a shift in microbial community composition, which buffers soil functionality against drought effects.

The results are novel because they compare field experiments with a long-term recurrent drought event to a single drought event. Although the methods used in the paper are standard and routine, such data from a unique experimental combination based on period of drought treatment are rare and therefore of interest to the community. Overall, I enjoyed reading the paper. I have some minor comments aimed at improving the paper.

AUTHOR RESPONSE: We thank the reviewer for the positive comments and feedback. We addressed each comment individually (in red for clarity) and integrate all the comments and suggestions in the manuscript (visible in red).

Conclusions are original, novel and well supported by the generated data. Functional predictions made using PICRUSt are weak but they fit well with the other more quantitative data. The functional genes identified are also on expected lines with no major surprises. The authors don't attribute the changes in community structure and function as well as other physiological indicators such as stoichiometry on changes in plant community and litter chemistry and root exudation which are highly likely after 10 years of drought. These only get a passing mention at the end of the discussion section.

AUTHOR RESPONSE: We agree with the reviewer about the importance of plants in the investigated processes. To address the reviewer's comment, we now included new paragraphs in the discussion section (see L. 532-540) to expand discussion on the possible role of the plant community and plant carbon inputs.

Work is convincing with plenty of data of different kinds supporting the overall story. The discussion section is very well written with relevant literature properly cited and discussed in the context of the study. Some more papers directly relevant to the discussion on drought adaptation of bacterial communities could be useful. There are clearly many soil relevant papers on drought and osmoprotection that are missing, which could enhance the discussion thus making it more credible. Instead of listing the papers here, I recommend the authors go through my recently published paper on microbial gene expression and metabolite response to a legacy of long-term drought: Malik, A.A., Swenson, T., Weihe, C. et al. Drought and plant litter chemistry alter microbial gene expression and metabolite production. ISME

J 14, 2236–2247 (2020). <https://doi.org/10.1038/s41396-020-0683-6> Here we also discuss the legacy effects of drought in plant community that changes the chemistry of plant organic matter which are resources for microbes.

AUTHOR RESPONSE: We thank the reviewer for the suggestion. We added multiple citations (see L.495-499) to strengthen the discussion section regarding microbial adaptation to drought, including:

Warren, C. R. Response of osmolytes in soil to drying and rewetting. *Soil Biol. Biochem.* **70**, 22–32 (2014).

Bouskill, N. J. *et al.* Belowground response to drought in a tropical forest soil. I. Changes in microbial functional potential and metabolism. *Front Microbiol* 7: 525. (2016).

Flemming, H.-C. *et al.* Biofilms: an emergent form of bacterial life. *Nat. Rev. Microbiol.* **14**, 563 (2016).

Malik, A. A. *et al.* Drought and plant litter chemistry alter microbial gene expression and metabolite production. *ISME J.* **14**, 2236–2247 (2020).

Around line 433, you mention “After a single drought event... the community was unable to cope with the drought stress”. This is supported by changes in biomass stoichiometry, but I wonder why it does not reduce microbial biomass, in fact there is a trend in the opposite direction.

AUTHOR RESPONSE: This is an interesting observation. Other studies have also shown a small increase or a null decrease in microbial biomass carbon. The reasons could be multiple, including accumulation of osmolytes and reserve compounds. We now added a sentence to briefly highlight this result (L. 467-469).

Information on incubation for growth rate and CUE measurements are not provided in as much detail as other methods. These are instead provided in supplementary section. The results of these analyses are also not presented in the main text, although these are quite valuable and add to the story.

AUTHOR RESPONSE: We now added these results in the main text (now Fig. 4). We also modified the method section accordingly (L. 610-613) and the other figure numbers. However, we maintained most of the information related to the methods in the supplementary section to comply with Nature Communications policies (methods should be less than 3000 words).

Line 334: not clear what is meant by “high functional complementary within bacterial phyla”. The authors could clarify it a bit.

AUTHOR RESPONSE: We meant to suggest that the phyla level is not sufficient to link identity to specific functions, which we mention in the discussion section already (now L. 501-503). Following the reviewer’s comment, we decided to remove that sentence.

Line 434: should be “change in biomass stoichiometry” – typo

AUTHOR RESPONSE: We corrected the typo. The new sentence reads: “changes in biomass stoichiometry” (L. 466).

460: “We found evidence of multiple genes connected to all these functions being enriched in the

bacterial community subjected to 10 years of recurrent drought. Consequently, being indicative of a drought adapted bacterial community.” Sounds like one whole sentence instead of two.

AUTHOR RESPONSE: We modified the phrase to make it one sentence as suggested by the reviewer (L.494).

Some of the paragraphs are very long. Consider splitting them based on sub-themes.

AUTHOR RESPONSE: We split some paragraphs in the discussion section (e.g. L. 436-438 and L.483-485) to address the reviewer’s comments.

Statistical analyses are appropriate and provided in sufficient detail. All data are either provided in the supplementary section or made publicly available in repositories.

All the best
Ashish Malik

Reviewer #2 (Remarks to the Author):

Canarini et al., investigated the effects of recurrent drought events and single drought events on microbial community and function. Their results showed a tendency of ecological memory of microbially mediated processes to a history of drought events. The authors further concluded that the ecological memory of drought in soil processes is caused by a shift in microbial community composition. However, I had some difficulties in understanding the causal relationship here. I am afraid that the author may over-interpret their results based on their experimental design and their data records. This is my major concern. In addition, I found several typos and grammar errors. I would suggest the authors double-check their writing before their resubmission.

AUTHOR RESPONSE: We thank the reviewer for the comments and feedback. We acknowledge the reviewer's concerns and to address them we modified the text to avoid over-interpretation of the link between microbial community composition and soil processes. We modified sentences in the abstract (L.35-40) and discussion (L.557 and L.562) sections. We also added a sentence (L. 545-548) to restate and clarify the concept of ecological memory and its dependence on legacy effects in the discussion section. We corrected the typos and checked the writing with the help of a native speaker.

Line 38-40. I am afraid of the over-interpretation here.

AUTHOR RESPONSE: We modified the sentence to tone down the interpretation and address the reviewer's comment (L.35-40).

Line 104 -119. I recognize the valuable contribution of long-term experiments. But, did you consider the climate and vegetation dynamics over the long term?

AUTHOR RESPONSE: We recognize that vegetation and climate are pivotal factors, and we addressed the reviewer's comments by expanding the discussion on these two subjects. Regarding vegetation dynamics, we now expanded the discussion section regarding possible contribution from the vegetation dynamics (and belowground inputs) to the observed processes (see L. 532-540). Regarding the climate dynamics we believe that by repeated sampling campaigns and by always maintaining a control subjected to the same climatic conditions, the possible influence of climate variation was reduced in our experiment. However, to further address this point raised by the reviewer we now added a sentence in the discussion section to highlight that climate variability represents another pivotal factor that might control drought responses (L. 520-521).

Fig. 1. What does dm in y-lab stand for? Why not normalizing per gram soil.

AUTHOR RESPONSE: The "dm" in figure one stands for "dry matter", as in soil dry matter, so this is already normalized per g of soil. We now changed it to "soil dm" to make it clearer and we applied the same change in Fig. 3.

Fig. 3. How did you classify enzymes related to the CNP cycle? Some enzymes are related to at least both of them.

AUTHOR RESPONSE: We classified the enzymes based on whether they are involved in the hydrolytic depolymerization of molecules that only contain organic carbon, or organic molecules that also contain N, P or S. As all organic molecules contain C, it is impossible to separate completely the acquisition of C, from the acquisition of nutrients such as N, S or P in organic molecules. We already clarified this at the bottom of Supplementary Table 3. To address the reviewer's comment, we added a phrase to clarify this point also in the Result section (L.249-254)

Fig. 5. Why not showing the values from control in the left panels?

AUTHOR RESPONSE: Values in Fig. 5 are obtained as a measure of the distance from the control (as explain in L.754-755). Therefore, the control represents the baseline. For this reason, the control values cannot be presented in this figure.

Line 515. I did not find related information in Supp Fig. 1.

AUTHOR RESPONSE: We apologise for the typo. It was meant to refer to Supplementary Fig. 15. We now corrected it in the text.

Line 516-517. How did you evaluate "grazed for a few days", if considering the substantial grazing effects on soil microbial processes?

AUTHOR RESPONSE: We now removed the sentence. We realized that this sentence might have created a misunderstanding. While the plots are part of a grassland site that is grazed, our experimental plots are excluded from grazing to avoid damage on the sensors and rain-out shelters.

Line 517. Will it be better to list the full name for FAO?

AUTHOR RESPONSE: We listed the full name as requested by the reviewer (L. 571)

Line 518-520. The dominant species are required.

AUTHOR RESPONSE: We added the names of dominant species (L. 573-574)

Line 520-521. In which year range?

AUTHOR RESPONSE: MAT and MAP was derived from a nearby climate station of the Hydrographic Services Tyrol, who provided a decadal data set (1988-1997) for the first site description by Bitterlich W & Cernusca A (1999) In: Cernusca A, Tappeiner U, Bayfield N (eds) Land-Use Changes in European Mountain Ecosystems. Blackwell Wissenschafts-Verlag, 368 p. For recent site-related references see the following papers, now cited in this article (L. 574):

Bahn, M., Knapp, M., Garajova, Z., Pfahringer, N. & Cernusca, A. Root respiration in temperate mountain grasslands differing in land use. *Glob. Chang. Biol.* **12**, 995–1006 (2006).

Bahn, M. *et al.* Soil respiration at mean annual temperature predicts annual total across vegetation types and biomes. *Biogeosciences* **7**, 2147 (2010)

Fuchslueger, L., Bahn, M., Fritz, K., Hasibeder, R. & Richter, A. Experimental drought reduces the transfer of recently fixed plant carbon to soil microbes and alters the bacterial community composition in a mountain meadow. *New Phytol.* **201**, 916–927 (2014).

Estiarte, M. *et al.* Few multiyear precipitation–reduction experiments find a shift in the productivity–precipitation relationship. *Glob. Chang. Biol.* **22**, 2570–2581 (2016).

Line 527. To make your study more comparable with others, you may calculate how much precipitation intercepted by rainout shelters?

AUTHOR RESPONSE: The annual precipitation reduction was already calculated in a previous study and we now cite the study and added the relevant information (annual precipitation reduction = 34%, now L.581-582)

Line 529. Be consistent with “rain-out” or “rainout” across your MS.

AUTHOR RESPONSE: We now corrected rain-out to rainout and consistently use it across the manuscript.

Line 676. For ANOVA and some other data analysis, data normality and residual distribution should be reported.

AUTHOR RESPONSE: We now added a Supplementary file (“Statistical report”) where plots of standardized residuals versus predicted values, frequency histogram and QQ-plot are reported for each variable used to test ANOVA. We also added a statement in the Methods section to indicate this (L. 739-741).

Reviewer #3 (Remarks to the Author):

Remarks to the authors:

Manuscript summary:

The rationale for this study is understanding the impacts of drought events (whose frequency is associated with climate change) and its effect on ecological processes in soils. The authors test the hypothesis that drought events can lead to “ecological memory formation”, which is defined as the capacity of past states to influence present or future responses. The focus of this study is the impacts of abiotic (edaphic) and biotic (microbial) factors into this ecological memory formation. Specifically, this effort focuses on a very unique experimental field site where ten years of simulated recurrent drought is compared to a single drought event (as well as other lengths of simulated drought events). The main finding of this study is that the ten year recurring drought response of the soil is different from a single drought event, which the authors posit can provide stability for soil functionality against drought effects. Further, this impact is mainly biological and not observed in abiotic measurements. The authors conclude that the shift in microbial community composition indicates an ecological memory of drought in soil processes.

This paper was well-written, and the introduction was helpful in setting the rationale for this study. In a brief literature search of drought microbial legacy effects, I found it justified that the impacts of drought on shifting microbial communities were well-studied and that this study was novel for its focus on specifically long-term recurrent drought events (> 1 year).

AUTHOR RESPONSE: We thank the reviewer for the positive comments and for the critical and constructive suggestions. We believe that the reviewers addressed important points and that these comments allowed us to further clarify and strengthen the manuscript. We addressed the reviewer comments at the end of the page (in red for clarity) and integrate all the comments and suggestions in the manuscript (visible in red).

The focal results in this effort in my opinion were the lack of differences observed in soil edaphic properties and the significant differences observed in both community structure and the enzyme potential. The observation of a shift between one year and ten year drought events were clear to me. My challenge in justifying the findings can be identified in the key discussion paragraph on Lines 492 and 503.

“Via a unique experimental setting and repeated sampling campaigns, we were able to disentangle effects of single drought vs recurrent drought events and assess ecological memory formation. We show two key findings: (i) recurrent drought strengthens shifts in the composition of both bacterial and fungal communities; (ii) soil multifunctionality is partially restored by recurrent drought events. We further suggest that microbial community shifts and changes in the microbially mediated cycling of carbon and nutrients, likely reflects community-dependent strategies to adapt to drought and controlling soil functions.”

I do not find the two key findings adequately justified. Mainly, I agree with the authors that the shift is present but the formation of “legacy” or “memory” is not clear to me. I do not see results that STRENGTHEN THE SHIFT in composition in the ten year samples. Figure 1 shows that the ten year is more similar to controls via microbial biomass but does not directly show a change in the magnitude of the shift, only that 1-year and 10-year are different. Figure 2 results also show differences between control, 1 year, and 10 year generally but do not in my opinion provide insights into the magnitude of this shift. And similar thoughts on the other figures. The authors bring in some other datasets in the Supplemental to include other drought events. Firstly, I found it difficult to synthesize this extra dataset within the context of the main paper, and feel like it has an opportunity to maybe address my concerns (if data can be obtained from these samples) but is not currently used to address the identification and quantification of the “legacy effect”.

Second, I am not convinced that soil functionality is restored and that the current data informs a “restoration”. Partially, this is due to the lack of clarity on what the legacy effect is, and partially I am not clear on if restoration to a “baseline” condition was part of this experiment. One can argue that soil “multifunctionality” is shown to return to somewhat baseline in Figure 3 but the text is not clear to me on what results justify this statement. Perhaps it is the identification of the inferred functions from PICRUST that are the justification. As the authors point out themselves, I think this should be done with extreme caution. If this is a key finding, I believe that functional assays be developed based on the hypotheses that can be developed from these findings to justify the conclusion. In a review of literature, I came across this effort (<https://www.frontiersin.org/articles/10.3389/fmicb.2018.00294/full>), which defines legacy response groups and suggest that it may provide an approach to authors if they find it helpful. There are likely other model strategies in other ecosystems besides soils to quantify the “legacy” effect, but I think this would help strengthen the findings beyond a shift to actual “memory”.

AUTHOR RESPONSE: We thank the reviewer to give us the opportunity to further clarify and address these points. First, as stated in lines 99-104, the objective of this manuscript was not to quantify legacy effects, but to assess the formation of an ecological memory in soil microbial processes and the presence of legacies in the soil microbial community. Nevertheless, to address the reviewer’s comment and strengthen this part of the manuscript, we now expanded the analysis following the reviewer’s suggestion. We adopted a similar analysis as in the manuscript suggested by the reviewer, where we individuated Legacy Response Groups (for 16S, ITS1 and ITS2 and AMF-targeted primers) based on our DESeq2 analysis. We now present a new Figure (Supplementary Fig. 12), and a new section in the Results (L 228-236), Methods (L. 736-737) and Supplementary methods. The new analysis enabled us to quantify microbial groups showing a legacy effect and their relative contribution to the entire microbial community.

Second, regarding soil functionality, we believe that some misunderstandings might have occurred, mainly by the use of the word “restore”, as this would imply that the functionality is restored to control levels once drought is ended. Instead, we measured multifunctionality during a subsequent drought event, and therefore it is not comparable to ambient moisture conditions (as the word restore might imply), but it represents the response during a subsequent drought event. We made the respective changes to the text, to properly address the reviewer’s comment and clarify the issue. Particularly we

changed the word “restore” and modified some related sentences (L.551-554). We further toned down our interpretations (and therefore justifications) on the link between microbial community shifts and soil functions (L. 35-40, L. 557 and L. 562).

REVIEWERS' COMMENTS

Reviewer #1 (Remarks to the Author):

I am happy with the changes done by the authors in light of comments from all three reviewers.

A minor point I have is linked to the section around L230: "We specifically selected these two treatments to disentangle possible adaptations due to previous drought history from simple responses to drought." It is not clear to me what this means. Adaptation is a term used for physiological changes that increase the fitness of individuals. How are the authors separating adaptations from simple responses? What are simple responses? I think the whole section with the new analysis on legacy groups from amplicon sequencing data is not really adding much to the existing story which was supported by strong evidence from quantitative measures. I'd leave it to the judgement of reviewer 3 on this matter who had issues with the terms legacy and memory.

Reviewer #2 (Remarks to the Author):

The authors did a good job on their revision. I have no other concerns. Ji Chen

Reviewer #3 (Remarks to the Author):

The authors have done a nice job responding to reviewer comments. I am satisfied with their rebuttal and justifications.

REVIEWERS' COMMENTS

Reviewer #1 (Remarks to the Author):

I am happy with the changes done by the authors in light of comments from all three reviewers.

A minor point I have is linked to the section around L230: "We specifically selected these two treatments to disentangle possible adaptations due to previous drought history from simple responses to drought." It is not clear to me what this means. Adaptation is a term used for physiological changes that increase the fitness of individuals. How are the authors separating adaptations from simple responses? What are simple responses? I think the whole section with the new analysis on legacy groups from amplicon sequencing data is not really adding much to the existing story which was supported by strong evidence from quantitative measures. I'd leave it to the judgement of reviewer 3 on this matter who had issues with the terms legacy and memory.

RESPONSE: We modified the sentence in accordance with the reviewer's and editor's comment, especially regarding the use of the word adaptation. The new sentence now reads: "We specifically selected these two treatments to disentangle long vs short-term effects of drought". We also agree with the reviewer that the new analysis does contribute fundamentally to the manuscript and for this reason the figure is in the supplementary section. We left the analysis in the manuscript, as it was requested by another reviewer. We will remove it if this is explicitly requested.